# Position: The Turing-Completeness of Autoregressive Transformers Relies Heavily on Context Management

Guanyu Cui [1]   Zhewei Wei [1]   Kun He [2]

## Abstract

Many works make the eye-catching claim that Transformers are Turing-complete. However, the literature often conflates two distinct settings: (i) a *fixed Transformer system* setting, in which a fixed autoregressive Transformer is coupled with a fixed context-management method to process inputs of different lengths step by step, and (ii) a *scaling-family* setting, in which a family of different models (with increasing context-window length or numerical precision) is used to handle different input lengths. Existing proofs of Transformer Turing-completeness are frequently established in setting (ii), whereas real-world LLM deployment and the standard notion of Turing-completeness correspond more naturally to setting (i). In this paper, we first formalize the fixed-system setting, thereby providing a concrete characterization of how real-world LLMs operate. We then argue that results proved in the scaling-family setting provide theoretically meaningful resource bounds but do not establish Turing-completeness, thereby clarifying a common misinterpretation of existing results. Finally, we show that different context-management methods can yield sharply different computational power, and we advocate the position that context management is a central component that critically determines the computational power of real-world autoregressive Transformers.

## 1. Introduction

The computational power of Transformer variants has been studied extensively, motivated by the striking capabilities of Transformer-based large language models (LLMs). A prominent line of work claims that Transformers are Turing-complete (Dehghani et al., 2019; Pérez et al., 2019; Bhattamishra et al., 2020; Giannou et al., 2023; Merrill & Sabharwal, 2024; Li et al., 2024; Roberts, 2024; Back De Luca & Fountoulakis, 2024; Malach, 2024; Nowak et al., 2024; Qiu et al., 2025; Hou et al., 2025; Li & Wang, 2025; Jiang et al., 2026; Schuurmans, 2023; Schuurmans et al., 2024). In other words, Transformers are expressive enough to compute any computable function. However, existing proofs typically rely on assumptions that are not fully justified for real-world LLMs, which correspondingly weakens the solidity of these Turing-completeness claims. This leaves a natural question: *Do the theoretical assumptions made in existing Turing-completeness claims actually apply to autoregressive Transformers as they are used in practice?*

In prior work, it is common to adopt assumptions such as allowing the Transformer context window to grow with the decoding step, or requiring the numerical precision of token embeddings to increase with the number of tokens processed so far. However, such assumptions do not characterize the capability of a single Transformer; rather, they effectively describe the computational power of a collection of different Transformers used at different input lengths. As a result, these works study an object that deviates from the single-Transformer setting we aim to understand, and therefore do not reflect how LLMs are used in practice. Motivated by this observation, we categorize existing works into two regimes depending on whether the Transformer is assumed to be fixed: a *fixed-system* regime and a *scaling-family* regime.

In the fixed-system regime, one fixes a single pretrained Transformer with a fixed context window $N$, fixed finite numerical precision, and fixed weights, as in real-world Transformer-based LLMs. Given an input token sequence, the model processes it autoregressively by placing tokens into the context window and appending the decoded token to the end of the current sequence until a stopping condition is reached. However, when the length of input or intermediate results exceeds $N$, the current sequence cannot be fed into the context window directly. In this case, autoregressive decoding must be coupled with a fixed context-management mechanism that determines which tokens are provided to the

[1]Gaoling School of Artificial Intelligence, Renmin University of China, Beijing, China [2]DEKE Lab, Renmin University of China, Beijing, China. Correspondence to: Zhewei Wei <zhewei@ruc.edu.cn>, Kun He <hekun2023@ruc.edu.cn>.

*Proceedings of the 43rd International Conference on Machine Learning*, Seoul, South Korea. PMLR 306, 2026. Copyright 2026 by the author(s).

context window at each step, thereby enabling the system to process arbitrarily long inputs.

On the other hand, in the scaling-family regime, one considers a family of models with different context-window length, numerical precision, or depth, and uses them to process inputs of different lengths. Many universality results fall into this second regime, even when they are informally interpreted as statements about the first. We summarize several common assumptions that, when adopted, move the analysis away from a single fixed pretrained model and implicitly place it in the scaling-family regime:

- **Scaling or unbounded window:** assuming that inputs of arbitrary length fit into the context window, or that during autoregressive decoding each token can attend to the entire input and all previously generated tokens.

- **Scaling or unbounded precision:** assuming internal representations require precision that grows with the input length, for example log-precision, or directly using unbounded rationals or real numbers.

In contrast to the above assumptions, a real-world Transformer has a fixed maximum context-window length and fixed internal numerical precision that does not vary with the input length. Hence, once these assumptions are adopted, the object of study is no longer a fixed real-world Transformer, but rather a family of Transformers, where different Transformers are used for sequences of different lengths.

**Position and Contributions.** We argue that theoretical studies of the computational power of Transformers should explicitly state their assumptions and distinguish scaling-family settings from realistic fixed-system settings. Scaling or unbounded context windows, as well as growing or unbounded numerical precision, change the object of study from a single deployed Transformer system to a family of models. For real-world LLMs, **the way context management, or more broadly the model's "harness", is coupled with the Transformer is not a peripheral implementation detail but a central component that can fundamentally change the induced computational model.**

We support this position through three contributions:

- We formalize a computational model for a fixed Transformer system. This provides a reference setting for discussing real-world autoregressive LLM deployment.

- We distinguish fixed-system and scaling-family regimes and explain how common assumptions such as scaling context windows and growing numerical precision should be interpreted. Results in the scaling-family regime are meaningful, but they do not establish Turing-completeness of a fixed deployed Transformer system.

- Under the fixed-system formalization, we provide simple derivations showing that summarization-style context management yields only constant-space computation, while appending-style methods achieve linear space; more sophisticated mechanisms can even be Turing-complete. These examples illustrate why context management is a central component of the induced computational model.

**Related Work.** Our work is primarily motivated by a blog post (Akhlaghpour, 2024) that informally argues that many Turing-completeness claims for Transformers involve implicit issues. However, the blog post is informal and presents these concerns by surveying related works in chronological order. In contrast, we provide an explicit formalization, clearly separate existing settings into the fixed-system and scaling-family regimes, and highlight a key observation largely absent from the blog discussion: how the choice of context manager can substantially affect the computational power of real-world LLM systems.

**Roadmap.** Section 2 reviews the background needed for our formalization and analysis. Section 3 defines the computational process of a fixed Transformer system $(T, D, C)$. Section 4 explains why interpreting scaling-family results as "autoregressive Transformers are Turing-complete" reflects a misconception, and surveys representative claims by identifying the scaling assumptions they rely on. Section 5 analyzes the computational power and separations of fixed Transformer systems under different context-management methods. Finally, Section 6 summarizes our conclusion and presents a call to action.

## 2. Preliminaries

This section reviews the background concepts that will be used later: (i) Turing machines (TMs), transducers, and complexity classes, (ii) Boolean circuits, and (iii) a minimal abstraction of Transformers and autoregressive decoding.

### 2.1. Notation

Let $\Sigma$ be a finite alphabet. A string over $\Sigma$ is an element of $\Sigma^* := \bigcup_{\ell=0}^{\infty} \Sigma^\ell$, where $\Sigma^0 = \{\epsilon\}$ and $\epsilon$ denotes the empty string. For a string $x$, let $|x|$ denote its length. For integers $1 \leq a \leq b \leq |x|$, we write $x_{a:b}$ for the substring $x_a \cdots x_b$. A language $L$ over $\Sigma$ is a set of strings, i.e., $L \subseteq \Sigma^*$. A partial function from $A$ to $B$, denoted $g : A \rightharpoonup B$, is a mapping that may be undefined for some $a \in A$.

### 2.2. TMs, Transducers, and Complexity Classes

**Turing machines** (TMs) are a classical computational model proposed by Alan Turing (Turing, 1937) to formalize the notion of computability. Informally, a (multi-tape) Turing machine consists of $k$ tapes, each extending infinitely

in one direction and divided into cells that store symbols. Each tape is equipped with a read-write head that can move left or right. Initially, the input is written on the tape(s) according to a fixed convention, and all other cells contain a blank symbol. At each step, the machine reads symbols, updates its state, writes symbols, and moves the heads according to a finite set of rules. A **transducer** is a machine that maps an input string to an output string; one convenient model is a TM with a read-only input tape, a work tape, and a write-only output tape. We use standard complexity classes such as $\mathsf{DTIME}(t(n))$ and $\mathsf{DSPACE}(s(n))$ for languages that can be decided deterministically in $O(t(n))$ time or $O(s(n))$ auxiliary space. We use $\mathsf{FDTIME}(t(n))$ and $\mathsf{FDSPACE}(s(n))$ for the corresponding function problems. When discussing space complexity, we can ignore the constant number of tapes, since any $k$-tape Turing machine can be simulated by a single-tape one using at most a factor-$k$ increase in space.

### 2.3. Boolean Circuits

A **Boolean circuit** is a computational model for a Boolean function $f : \{0,1\}^n \to \{0,1\}^m$. It can be represented as a finite directed acyclic graph (DAG) with $n$ source nodes $x_1, \cdots, x_n$ as inputs and $m$ sink nodes $y_1, \cdots, y_m$ as outputs. All other nodes are logic gates such as AND, OR, NOT, and edges are wires carrying bits. Given an input assignment $x \in \{0,1\}^n$ to the input nodes, each gate is evaluated by applying its operation to the values on its incoming wires, and the values at the output nodes define $f(x) \in \{0,1\}^m$. Unlike a single Turing machine which can process inputs of arbitrary length, a single Boolean circuit only handles inputs of a fixed length. Therefore, in circuit complexity, we typically study a **family** of circuits $\{C_n : n \in \mathbb{N}\}$, where $C_n$ handles inputs of length $n$.

### 2.4. Turing-Completeness

**Turing-completeness** is a notion used to characterize whether a class of objects is at least as expressive as Turing machines. To define Turing-completeness, we first define Turing-computable functions. A partial function $f : \Sigma^* \rightharpoonup \Sigma^*$ is **Turing-computable** if there exists a TM $M$ such that for all $x, y \in \Sigma^*$, $f(x) = y$ if and only if $M$ halts on input $x$ and outputs $y$. A collection of objects $\mathcal{N}$ is **Turing-complete** if for every Turing-computable partial function $f$ there exists an object $N \in \mathcal{N}$ that computes $f$. From this definition, a necessary condition for Turing-completeness is that each object supports inputs of unbounded length.

### 2.5. Transformers

The Transformer (Vaswani et al., 2017) model is a well-known neural network architecture designed for sequence

modeling which relies primarily on self-attention mechanism to capture the content of the sentence and to generate results. Transformer models can be broadly categorized by whether they include encoder layers and/or decoder layers into three families: encoder-only Transformers (e.g., BERT (Devlin et al., 2019), RoBERTa (Liu et al., 2019)), encoder-decoder Transformers (e.g., the vanilla Transformer (Vaswani et al., 2017), T5 (Raffel et al., 2020), BART (Lewis et al., 2020a)), and decoder-only Transformers (e.g., GPT (Brown et al., 2020; Achiam et al., 2023), LLaMA (Touvron et al., 2023), Gemini (Gemini Team et al., 2023; 2024), DeepSeek (DeepSeek-AI et al., 2024; Guo et al., 2025), and Qwen (Bai et al., 2023; Yang et al., 2025)).

Given an input sentence, a tokenizer is applied before the Transformer to map the raw text to a sequence of discrete tokens from a finite vocabulary $\Sigma$. This token sequence is then fed into the Transformer. In practice, tokens are mapped to token IDs before being input to the Transformer. Since this is only an encoding choice, we still treat $\Sigma$ as the token domain. A pretrained decoder-only Transformer with context window length $N$ and fixed numerical precision can be abstracted as a fixed function $T$ that maps an $N$-token sequence[1] to next-token logits, and hence to a distribution over $\Sigma$. Formally, $T : \Sigma^N \to \Delta(\Sigma)$, where $\Delta(\Sigma)$ denotes the set of all distributions on $\Sigma$. As a supplement, many Transformer architectures use positional encodings to incorporate token-position information, either at the embedding stage (e.g., the sinusoidal PE in the vanilla Transformer (Vaswani et al., 2017) or the learned PE in BERT (Devlin et al., 2019)) or within self-attention (e.g., RoPE (Su et al., 2024)). In this case, we may regard the input within the context window as a length-$N$ string over $\Sigma \times \mathcal{P}$, where $\mathcal{P} = \{0,1\}^{p_{\mathrm{pos}}}$ for a fixed $p_{\mathrm{pos}}$. Thus, using positional encodings only enlarges the alphabet by a constant factor, and we will not mention them separately in what follows. For our purposes, the architectural details of attention are not essential and are deferred to Appendix A. What matters is that (i) the input length (the context-window length) of a single Transformer is a fixed constant $N$, and (ii) all parameters and internal representations use fixed finite precision.

### 2.6. Autoregressive Decoding and Context Management

Decoder-only Transformers are typically used via autoregressive decoding. Given a current token sequence $x = x_1 \cdots x_\ell$, we feed $x$ into the Transformer $T$ to obtain a next-token distribution $T(x)$, apply a decoding rule $D$ to select the next token $x_{\ell+1}$, and append it to the end of $x$ to form the new sequence, iterating this procedure. This process is

---

[1]Although the Transformer's input length is fixed to $N$, in practice a single forward pass can process any input sequence of length at most $N$. If the input is shorter than $N$, it is padded with `<pad>`. When we say that a sequence of length $< N$ is placed into the context window, we omit the padding step for brevity.

well defined when the sequence length does not exceed the context-window length $N$. To continue decoding when the sequence length exceeds $N$, one must introduce a context-management mechanism $C$ that determines which tokens are provided to the context window at each step. Section 3 formalizes this computation process as a fixed system.

# 3. The Fixed-System Regime for Transformers

We now formalize computation of autoregressive Transformers in the fixed-system regime. The goal is to make explicit what is fixed and what can grow with the decoding steps.

### 3.1. The System $(T, D, C)$ and Its Execution

Fix a finite token vocabulary $\Sigma$ and a context window length $N$. A *fixed Transformer system* is a triple $(T, D, C)$:

- $T$ is a fixed pretrained Transformer, viewed as a function that maps any string $w \in \Sigma^N$ in its context window to a distribution $\mu_w \in \Delta(\Sigma)$ over the next token.

- $D$ is a deterministic[2] constant-time decoding rule that maps a distribution over $\Sigma$ to an output token in $\Sigma$. Typical choices include greedy decoding ($\arg\max$) or a fixed-temperature softmax sampling with a fixed seed.

- $C$ is a deterministic context manager. At step $t$, it maintains internal state and strings $r^{(t)} \in \Sigma^*$, produces a context string $w^{(t)} \in \Sigma^N$ via $w^{(t)} = C_w\left(r^{(t)}\right)$, and then updates its state and strings after decoding $\hat{x}_{t+1}$ by setting $r^{(t+1)} = C_r\left(\hat{x}_{t+1}, r^{(t)}\right) \in \Sigma^*$.

Given an input string $x = x_1 x_2 \cdots x_n \in \Sigma^*$, we feed it to $C$ and set $r^{(1)} = x$. The system then repeatedly applies $w^{(t)} = C_w\left(r^{(t)}\right)$ to form the token sequence fed into the context window, $\hat{x}_{t+1} = D\left(T\left(w^{(t)}\right)\right)$ to obtain the next token, and $r^{(t+1)} = C_r\left(\hat{x}_{t+1}, r^{(t)}\right)$ to update its maintained strings, until a stopping condition of $C$ is met. The system output is the generated token sequence (or a final answer extracted by a fixed convention). For decision problems, we can designate special accept and reject tokens and say that the system decides a language by eventually emitting exactly one of them. For function computation, we view the system as a transducer that outputs a string.

### 3.2. Some Common Context-Management Methods

We summarize several common context-management methods that will be referenced later.

---

[2]For simplicity, we assume the next-token selection rule is deterministic and single-valued. If not, one can adopt a nondeterministic-TM-style convention and define the Transformer's output as the set of all valid next tokens.

**Summarization-Style.** A common approach is to compress earlier parts of the history via summarization, e.g., latent compression methods such as AutoCompressor (Chevalier et al., 2023) and ICAE (Ge et al., 2024), or the `/compress` or `/compact` commands as used in Gemini-CLI (Google, 2026a), OpenAI Codex CLI (OpenAI, 2026a), and Claude Code (Anthropic, 2026a), which replaces part of the earlier context with a shorter summary. Given an input sequence $x = x_1 \cdots x_n \in \Sigma^*$ and setting $r^{(1)} = x$, an example summarization-style context-management process operates as follows:

- **Normal decoding phase.** If $|r^{(t)}| < N - 1$, the context manager provides $w^{(t)} = r^{(t)}$ to the Transformer. Given the decoded next token $\hat{x}_{t+1}$, there are two cases. If $\hat{x}_{t+1}$ is the end-of-sequence token `<EOS>`, the system terminates. Otherwise, set $r^{(t+1)} = r^{(t)} \circ \hat{x}_{t+1}$.

- **Summarization phase.** If $|r^{(t)}| \geq N - 1$, the context manager provides $w^{(t)} = \texttt{} \circ r^{(t)}_{1:N-1}$ to the Transformer, where `` is a summary-instruction token that triggers summarization. The Transformer then decodes $\hat{x}_{t+1}$ as a summary of the contents in the window, and we update $r^{(t+1)} = \hat{x}_{t+1} \circ r^{(t)}_{N:|r^{(t)}|}$. For simplicity, we assume that the summary is a single token. To generate a multi-token summary, we can reserve a budget of $t$ tokens (with $1 < t \leq N/2$) in the context window, that is, trigger summarization when $|r^{(t)}| \geq N - t$ instead of $N - 1$. Once summarization is triggered, the model can generate a length-$< t$ summary autoregressively (i.e., by invoking the normal decoding phase as a subroutine) and prepend the resulting summary sequence to $r^{(t)}$.

Figure 1 illustrates the summarization-style context manager described above.

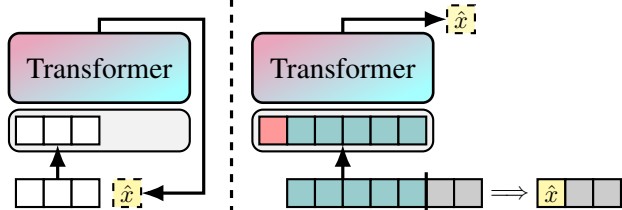

*Figure 1.* Illustration of summarization-style context management. Left: normal decoding phase. Right: summarization phase. Teal tokens indicate the segment to be summarized, the red token is the summary-instruction token ``, and the yellow token denotes the generated summary $\hat{x}$.

**Appending-Style.** Another method, similar to that used by Schuurmans et al. (Schuurmans et al., 2024), treats the Transformer's context window as a sliding window and appends predicted tokens to the end of the sequence. Given an input sequence $x = x_1 \cdots x_n \in \Sigma^*$ and setting $r^{(1)} = x$, an example appending-style context-management process

operates as follows. At step $t$, the context manager provides $w^{(t)} = r_{1:\min\left\{|r^{(t)}|, N\right\}}^{(t)}$ to the Transformer and obtains $\hat{x}_{t+1}$. If $\hat{x}_{t+1}$ is a halting token in a designated set $H$, the system terminates. Otherwise, it appends the decoded token to the end of $r^{(t)}$ and shifts the window forward by one token, i.e., $r^{(t+1)} = r_{2:|r^{(t)}|}^{(t)} \circ \hat{x}_{t+1}$. Figure 2 illustrates the appending-style context manager described above.

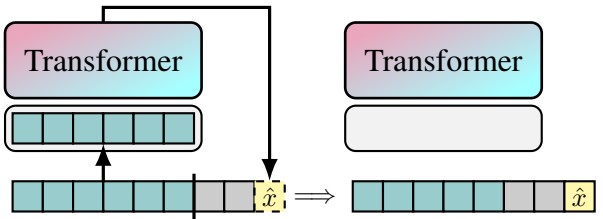

*Figure 2.* Illustration of appending-style context management. Left: decode a new token $\hat{x}$ to append to the end of the sequence. Right: shift the window forward by 1 token.

**Other Methods and Remark on the Power of $C$.** Beyond the two methods mentioned above, many other context-management mechanisms have been proposed. For example, many systems write important information to external storage and retrieve it when needed, as in retrieval-augmented generation and external-memory-based agents (Guu et al., 2020; Lewis et al., 2020b; Packer et al., 2023). Other systems, such as ToolFormer (Schick et al., 2023), GPT (OpenAI, 2026b), Claude (Anthropic, 2026b), and Gemini (Google, 2026b), also incorporate tool calls or function calls to access external resources. We remark that the definition of a context manager $C$ is very broad. With such an unrestricted $C$, the overall system can become trivially powerful. For example, if $C$ can call a Turing-complete tool (e.g., by executing Python programs), then $(T, D, C)$ is Turing-complete regardless of $T$. Therefore, when we analyze several of these methods in Section 5, we explicitly restrict the capabilities of $C$, while still covering common deployed mechanisms such as summarization and sliding windows.

## 4. Alternative Views

Many works make (or are commonly interpreted as making) the claim: *Autoregressive Transformers are Turing-complete.* To interpret this claim precisely, one must specify the computational regime. In the fixed-system regime, Turing-completeness would mean that for any Turing machine, there exists a single fixed system $(T, D, C)$ can correctly simulate it on inputs of arbitrary length. In contrast, in the scaling-family regime, it suffices to prove that for each input length $n$ (or number of steps $t$), there exists a Transformer model whose context-window length or numerical precision depends on $n$ (or $t$) that simulates the desired computation. Only the fixed-system regime matches the way

real-world LLMs are deployed, where the pretrained model is fixed and inputs are handled by a fixed context-manager.

In this section, we first clarify why Turing machine simulation results proved in the scaling-family regime should not be interpreted as establishing Turing-completeness. We then survey representative works and identify whether they implicitly adopt the assumptions listed in Section 1, which place their analysis in the scaling-family regime and therefore do not by themselves establish Turing-completeness of a fixed autoregressive Transformer system.

### 4.1. Scaling Models Does Not Mean Turing-Complete

By Definition 2.4, to establish Turing-completeness for autoregressive Transformers in the fixed-system regime, one must show that for every Turing-computable partial function $f$ there exists a single system $(T, D, C)$ that computes $f$ on inputs of unbounded length. In contrast, showing that longer inputs or longer decoding runs can be simulated by a scaled model only yields a resource bound for Transformers to handle different input lengths. This is closer in spirit to a circuit-complexity result. Savage (Savage, 1972) showed that if a language $L$ is decidable in time $\mathsf{DTIME}(T(n))$, then for each $n$ there exists a circuit $C_n$ of size $O((T(n))^2)$ that decides $L$ on length-$n$ inputs. Scaling circuit size with $n$ does not make a single circuit universal. It produces a family of circuits, one for each input length. Similarly, scaling the context-window length or the numerical precision yields a scaling-family viewpoint rather than Turing-completeness of a fixed pretrained Transformer system. Beyond this conceptual misalignment, real-world LLMs operate with a fixed finite context window and rely on context management to handle inputs of arbitrary length, whereas the scaling-family regime assumes that arbitrarily long histories can be placed into the context window. This also departs from how LLMs are used in practice.

### 4.2. Works That Scale Transformers

In this subsection, we review representative works that claim Turing-completeness for Transformers, or claim that Turing machines can be simulated by Transformers, and identify which ones explicitly or implicitly fall into the scaling-family regime. We group these works into two categories.

**Group A: Scaling Window.** A large fraction of "Transformer simulates TM" arguments implicitly assume that, given an input sequence of length $n$, at decoding step $t$ the token at position $n + t$ can attend to all tokens in the sequence within a single Transformer forward pass. This effectively requires the context-window length at step $t$ to be at least $n + t$. Works that explicitly or implicitly adopt this assumption include (Dehghani et al., 2019; Pérez et al., 2019; Bhattamishra et al., 2020; Merrill & Sabharwal, 2024;

Li et al., 2024; Roberts, 2024; Malach, 2024; Nowak et al., 2024; Qiu et al., 2025; Hou et al., 2025; Jiang et al., 2026). This group also includes looped or recurrent Transformer variants that repeatedly modify a fixed length input. Although the context-window length may not grow with the number of decoding steps in these formulations, handling longer inputs still requires a model with a larger context window. Thus, these works are also scaling the window in essence. Works that explicitly or implicitly adopt this assumption include (Giannou et al., 2023; Back De Luca & Fountoulakis, 2024; Li & Wang, 2025).

**Group B: Scaling Precision.** Several constructions require the precision of token embeddings or internal representation matrices to grow with the input length $n$ or the number of decoding steps $t$, or they assume exact unbounded-precision real or rational arithmetic. This is incompatible with the fixed finite precision of deployed models. Works that explicitly or implicitly adopt this assumption include (Dehghani et al., 2019; Pérez et al., 2019; Bhattamishra et al., 2020; Giannou et al., 2023; Merrill & Sabharwal, 2024; Li et al., 2024; Roberts, 2024; Back De Luca & Fountoulakis, 2024; Nowak et al., 2024; Qiu et al., 2025; Hou et al., 2025; Jiang et al., 2026).

Table 1 summarizes the assumptions used in related works. Regarding the context-window length, all related works in the scaling-family setting assume a non-constant context window. Regarding numerical precision, only two works assume constant precision. These misalignments make the corresponding Turing-completeness claims more subtle and arguably debatable.

### 4.3. Works with a Fixed Transformer

Two works highlight that, even with fixed numerical precision and a fixed context window, computational universality can arise by modifying the overall system, especially the context-management interface.

1. Schuurmans (Schuurmans, 2023) claims that a fixed Transformer-based LLM will be Turing-complete when augmented with access to external memory. The paper provides experimental evidence that a real LLM can simulate all single-step operations of a Turing-complete language, but it does not give a constructive theoretical proof of how a Transformer realizes these operations.

2. Schuurmans et al. (Schuurmans et al., 2024) propose an extended autoregressive decoding system with a fixed context window that can be Turing-complete by generating up to two output tokens per step and appending them to the token sequence for future processing.

These works suggest that, in the fixed Transformer system

*Table 1.* Assumptions used in representative works. "Window" denotes the context-window length, i.e., the sequence length processed by a single self-attention operation. "Precision" denotes the number of bits used to represent one token. Here $n$ denotes the initial input length, $t$ denotes the number of decoding steps (often interpreted as the CoT length), and $s(n)$ denotes the space complexity to recognize the language. "Unbounded" precision means the proof assumes exact real / rational computation without a concrete precision bound.

| Window | Precision | Work |
|---|---|---|
| $n + t$ | unbounded | (Dehghani et al., 2019),[*] (Pérez et al., 2019; Bhattamishra et al., 2020; Roberts, 2024; Nowak et al., 2024; Jiang et al., 2026) |
| | $\mathrm{poly}(n)$ | (Li et al., 2024)[†] |
| | $O(\log(n+t))$ | (Merrill & Sabharwal, 2024; Li et al., 2024),[†] (Qiu et al., 2025), (Hou et al., 2025)[*] |
| | $O(1)$ | (Malach, 2024)[*†] |
| $n$ | unbounded | (Back De Luca & Fountoulakis, 2024) |
| | $O(\log n)$ | (Giannou et al., 2023) |
| $s(n)$ | $O(1)$ | (Li & Wang, 2025) |

[*] The original paper does not explicitly state an upper bound on the numerical precision, to the best of our understanding.

[†] The original paper does not explicitly claim that Turing-completeness has been proved; instead, it claims that one can construct a Transformer that simulates the execution of a Turing machine on a given input instance.

regime, Turing-completeness may be achieved by modifying the decoding procedure or designing different context-management mechanisms, so scaling the Transformer itself is not necessary. Section 5 further analyzes how different context-management methods affect the computational power of a fixed Transformer system.

## 5. Our Position: Context Management Matters for Autoregressive Transformers

In Section 4, we point out that existing studies often conflate settings that align with real-world Transformers with settings that depart from them, while broadly claiming that Transformers are Turing-complete. Our position is different. We argue that, in the fixed Transformer system setting that better reflects real-world deployed Transformers, the method of context management has a greater impact on the system's computational power than the Transformer itself.

To support our view, we use simple theoretical derivations to analyze how different context-management methods affect the upper bound of the computational power of the entire system. Specifically, we examine two context-management methods, namely summarization-style context management and appending-style context management. We prove that, when combined with a fixed-context-window, fixed-precision Transformer, summarization-style context management leads to a system bounded by a constant-space Turing machine, whereas appending-style context management yields a system with the same power as a linear-space Turing machine. In addition, drawing on existing works, we point out that the resulting systems are Turing-complete if the Transformer is allowed to read from and write to memory through a context-management mechanism, or if the Transformer is allowed to decode one or two tokens at a time.

Through the survey in Section 4.2 and Section 4.3, we show that most papers focus only on the importance of the Transformer itself while overlooking the importance of harnesses such as context management. Therefore, our position paper highlights an issue that has been overlooked by the community and provides corrective guidance for subsequent research in this direction by emphasizing the importance of context management.

### 5.1. Analysis of Different Context-Management Methods

Next, we discuss how different context-management methods affect the computational power of Transformer systems with a fixed context-window size and fixed precision. As discussed in Section 3, an unrestricted context manager $C$ can make the overall system trivially powerful. Accordingly, we focus on "simple enough" context managers. Such a manager maintains only $N$ memory cells for the Transformer's context window and $O(1)$ auxiliary memory cells for states, beyond storing token strings organized in simple data structures such as queues. It can apply only fixed local operations to these maintained strings, for example extracting a prefix by popping a constant number of tokens, shifting a window by a constant amount, appending tokens by pushing them to the tail, and querying the Transformer for the next token. This convention captures the summarization-style and appending-style context managers described in Section 3.2, while excluding the ability to run general algorithms.

**Summarization-Style Context Management.** Next, we show that the system induced by summarization-style context management is upper bounded by a deterministic constant-space Turing machine, or $\mathsf{FDSPACE}(1)$[3].

---

[3]Strictly speaking, space complexity classes are defined for machines that halt on all inputs. A machine using bounded space need not halt, so our use of space-complexity notation is a mild

**Proposition 5.1.** *Any fixed Transformer system $(T, D, C)$, where $C$ is a summarization-style context manager, can be simulated by a Turing machine using constant space.*

*Proof.* Assume $T$ has context-window length $N = O(1)$ tokens, with each input token in $\Sigma$. An observation that will be used implicitly in what follows is that a single decoding step of a Transformer, $D(T(\cdot)) : \Sigma^N \to \Sigma$, can be simulated by a Turing machine in constant space (with the constant depending on $N$).

We construct a one-way transducer to simulate each step $t$ of the computation process of the system $(T, D, C)$, where the head of the read-only input tape never moves left (recall that a transducer can be formalized as a three-tape Turing machine with a read-only input tape, a read/write work tape, and a write-only output tape).

*Initialization.* At initialization, the input tape contains the input string $x = x_1 \cdots x_n \in \Sigma^*$, the other two tapes are blank, and all heads start at the first cell. We write a boundary marker $\#$ in cell $N + 1$ of the work tape to partition the tape into two parts: cells $1$ through $N$ simulate the contents of the Transformer's context window $w^{(t)}$, while cell $N + 2$ onward is the workspace region used to simulate a single Transformer decoding step $D(T(w^{(t)}))$.

*Simulating step $t$ of the system.* At the beginning of the simulation of the $t$-th step of a Transformer system with summarization-style context management, we copy (token by token) the symbol under the input head and append it to the end of the token string on the work tape, until either the number of tokens on the work tape reaches $N - 1$ or the input tape is exhausted. We then distinguish two cases based on the number of tokens currently on the work tape:

- If the number of tokens on the work tape is less than $N - 1$ (denote the length by $\ell < N - 1$), we simulate one Transformer decoding step $D(T(\cdot))$ in the workspace region using these tokens as input. We then write the decoded token $\hat{x}_{t+1}$ into cell $\ell + 1$, and finally clear the workspace region.

- If the number of tokens on the work tape is $N - 1$, we shift these tokens one cell to the right and write the summary token `` in the first cell. We then simulate one Transformer decoding step $D(T(\cdot))$ in the workspace using these length-$N$ token sequence, write the decoded token $\hat{x}_{t+1}$ into the first cell of the work tape, and finally clear the work tape so that only $\hat{x}_{t+1}$ in the first cell and the boundary marker $\#$ in cell $N + 1$ remain.

The transducer then proceeds to simulate step $t + 1$ of the system. It is easy to see that, during the simulation of any

---

abuse adopted for notational convenience. The same caveat applies throughout.

step $t$, the token sequence on the work tape (excluding the boundary marker #), concatenated with the token sequence on the input tape at and to the right of the input head, is always equal to $r^{(t)}$. The first case, in which placing all of $r^{(t)}$ into the context window yields a length no greater than $N - 1$, corresponds exactly to $|r^{(t)}| < N - 1$, while the second case corresponds to $|r^{(t)}| \geq N - 1$. In both cases, the Turing machine simulates the Transformer system correctly. Consequently, the above Turing machine correctly simulates the behavior of $(T, D, C)$. For the space bound, the length of the work-tape portion of $r^{(t)}$ is at most $N$, and the simulation additionally requires constant space to simulate one Transformer decoding step. Hence, the total space usage is constant. □

A standard fact of regular languages is that $\mathsf{REG} = \mathsf{DSPACE}(1) = \mathsf{NSPACE}(1)$ (see, e.g., (Gasarch, 2015; Rothvoss, 2024)), where $\mathsf{REG}$ denotes the class of regular languages, and $\mathsf{NSPACE}(1)$ denotes constant space on a nondeterministic Turing machine. Therefore, when $C$ is summarization-style and the Transformer's context-window length, depth, and numerical precision are fixed constants (i.e., do not scale with the input length $n$), the system $(T, D, C)$ can recognize only regular languages (assuming it halts), even if decoding is non-deterministic. In particular, such a fixed system cannot recognize non-regular languages such as equality $\{x\#x : x \in \Sigma^*\}$, palindromes $\{x\#x^R : x \in \Sigma^*\}$, or binary addition $\{\mathrm{bin}(x)\#\mathrm{bin}(y)\#\mathrm{bin}(z) : x + y = z\}$.

Another point worth noting is that in our construction, the workspace uses space $s$ that is at least of the same order as $N$ (with an additional contribution coming from the space needed to simulate the Transformer). When translating this into an equivalent finite-state automaton, the resulting upper bound on the number of states grows exponentially in $s$, which in a sense provides an alternative perspective on scaling laws (Kaplan et al., 2020). Although this does not change the theoretical fact that the decidable languages remain regular, the languages the system can decide become more complex from the perspective of the number of states required by the corresponding finite-state automaton.

**Appending-Style Context Management.** Next, we discuss the computational power of appending-style context management.

**Proposition 5.2.** *Any fixed Transformer system $(T, D, C)$, where $C$ is an appending-style context manager, can be simulated by a deterministic Turing machine using space linear in the input length $n$.*

*Proof.* The construction is straightforward. At initialization, the Turing machine copies the entire current sequence to the work tape. At decoding step $t$, it copies the first $N$ tokens

into a workspace region starting from cell $n + 1$, simulates one Transformer decoding step, and obtains $\hat{x}_{t+1}$. It writes $\hat{x}_{t+1}$ into cell $n + 1$, shifts the content in cells 2 through $n + 1$ one position to the left, and clears all other cells on the work tape. □

The proposition above shows that any fixed Transformer system with appending-style context management can be simulated by a linear-space Turing machine. We can also show the converse direction. Before stating the proposition and its proof, we briefly review the extended autoregressive decoding system defined by Schuurmans et al. (Schuurmans et al., 2024). Their $N$-gram extended autoregressive decoding is based on a sliding window. Initially, the current token string is set to the input $x = x_1 \cdots x_n$. At each decoding step, the first $N$ tokens are fed into a fixed decoding function $M : \Sigma^N \to \Sigma^*$, which outputs a decoded token string (and the process halts if the output falls in a designated halting set). The decoded string is appended to the end of the current sequence, and the first token is removed. When the output length of $M$ is at most $K$, the resulting system is called an $(N, K)$-restricted system. We also need the following lemma (Lemma 5.3), whose proof is deferred to Appendix B.

**Lemma 5.3.** *Let $\Sigma$ be a finite alphabet. For any function $f : \Sigma^2 \to \Sigma$, there exists a decoder-only Transformer $T$ such that for every length-2 context $(a, b) \in \Sigma^2$, greedy decoding outputs $f(a, b)$.*

We are now ready to state and prove the converse direction.

**Proposition 5.4.** *Any deterministic Turing machine that uses linear space can be simulated by a fixed Transformer system $(T, D, C)$, where $T$ has context-window length 2 and $C$ is an appending-style context manager.*

*Proof.* By the definition of an $(N, K)$-restricted system, our fixed Transformer system with an appending-style context manager is a special case with $K = 1$, where the decoding function $M$ is implemented by the composition of a Transformer $T$ and a decoding rule $D$[4]. Schuurmans et al. (Schuurmans et al., 2024) show that the computation of any linear-space Turing machine can be simulated by a $(2, 1)$-restricted system. According to Lemma 5.3, any function $f : \Sigma^2 \to \Sigma$ can be realized by a Transformer $T$ with context-window length 2 coupled with the greedy decoding rule, and the claim follows. □

Propositions 5.2 and 5.4 suggest that a single fixed Transformer system with appending-style context management

---

[4]In the definition of (Schuurmans et al., 2024), when $K = 1$ the decoding output of $M$ can contain zero or one token. In our appending-style context management, this can be modeled by including an empty symbol in $\Sigma$ and ignoring it when appending.

has the same computational power as a linear-space Turing machine. In particular, it can decide exactly the class of deterministic context-sensitive languages, DCSL $=$ DSPACE$(n)$ (see, e.g., (Ibarra, 1991; Otto, 2006)). Schuurmans et al. (Schuurmans et al., 2024) also show that a $(2, 2)$-restricted system is Turing-complete. If one could extend Lemma 5.3 to show that a context-window-2 Transformer can realize any function $\Sigma^2 \to \Sigma^2$ by applying greedy decoding to the two output positions $Y(1, :)$ and $Y(2, :)$, then this would yield a Turing-complete system as well. Note that this changes the decoding interface from next-token decoding to next-two-tokens decoding. In Appendix C, we briefly discuss how one might preserve next-token decoding by modifying the context manager to perform two decoding steps and then discard one of the produced tokens.

Table 2 summarizes the computational power of a fixed Transformer system under different context managers. We conclude that different context managers can endow the same underlying model with markedly different computational power, which supports our position: **the Turing-completeness of real-world autoregressive Transformers relies heavily on context management**.

*Table 2.* Computational power under different context managers. Here $\equiv$ denotes equivalence and $\leq$ denotes no more powerful than.

| Work | Context Management | Power |
|---|---|---|
| (Schuurmans, 2023) | read / write memory | $\equiv$ TM |
| (Schuurmans et al., 2024) | $(2, 2)$-restricted sys. | $\equiv$ TM |
| (Schuurmans et al., 2024) | $(2, 1)$-restricted sys. | $\equiv O(n)$-space TM |
| Ours | appending-style | $\equiv O(n)$-space TM |
| Ours | summarization-style | $\leq O(1)$-space TM |

## 6. Summary and Call to Action

In this paper, we formalized the computation of a fixed Transformer system and distinguished two settings for the Turing-completeness of autoregressive Transformers. We showed that the scaling-family setting is theoretically misaligned with the notion of Turing-completeness and, in practice, does not match how real-world LLMs are deployed and used. We then analyzed the fixed-system setting and demonstrated that context management can substantially affect the computational power of Transformers.

We conclude with three calls to action.

- First, claims about Transformer Turing-completeness should **explicitly state their computational setting and assumptions**. Scaling-family results are valuable for understanding resource requirements, such as context length, precision, and depth, but should not be interpreted as Turing-completeness of a fixed real-world Transformer system.

- Second, since real-world deployed LLM systems consist of a fixed-context-window, fixed-precision Transformer together with a particular context-management method, theoretical work should **devote more attention to the capabilities of the overall system obtained by combining different context-management methods with a fixed Transformer**.

- Third, theoretical works should **look beyond qualitative Turing-completeness analyses under idealized assumptions, and complement them with capability claims stated in terms of explicit resource budgets and learnability criteria**, since Turing-completeness only concerns whether a function is computable under a specified encoding, and does not by itself answer whether the model can acquire, generalize, or robustly use the relevant solution.

## Acknowledgments

This research was supported in part by National Natural Science Foundation of China (No. L2524018, No. U2241212, No. 92470128, No. 62472430) and by Industrial AI Solutions, Li Auto Inc. We also wish to acknowledge the support provided by the fund for building world-class universities (disciplines) of Renmin University of China, by Engineering Research Center of Next-Generation Intelligent Search and Recommendation, Ministry of Education, by Intelligent Social Governance Interdisciplinary Platform, Major Innovation & Planning Interdisciplinary Platform for the "Double-First Class" Initiative, Public Policy and Decision-making Research Lab, and Public Computing Cloud, Renmin University of China.

The work was partially done at Gaoling School of Artificial Intelligence, Beijing Key Laboratory of Research on Large Models and Intelligent Governance, Engineering Research Center of Next-Generation Intelligent Search and Recommendation, MOE, and Pazhou Laboratory (Huangpu), Guangzhou, Guangdong 510555, China.

We would also like to thank Hessameddin Akhlaghpour for the blog post *Are Transformers Turing-Complete? A Good Disguise Is All You Need*, which inspired us to further examine and carefully distinguish the assumptions in existing theoretical works that align with real-world practice from those that do not. It also motivated us to investigate, under assumptions that better reflect real-world Transformers, how the capabilities of these models are closely tied to context management. We also thank the anonymous reviewers for their valuable comments.

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

# Appendices

## A. More Details of Transformers

We first introduce our notation for vectors and matrices. Bold lowercase letters such as $\boldsymbol{x}$ denote vectors (column vectors unless stated otherwise), and bold uppercase letters such as $\boldsymbol{X}$ denote matrices. We write $\boldsymbol{x}(i)$ for the $i$-th entry of $\boldsymbol{x}$. For a matrix $\boldsymbol{X}$, $\boldsymbol{X}(i,:)$ denotes the $i$-th row (as a row vector), $\boldsymbol{X}(:,j)$ denotes the $j$-th column (as a column vector), and $\boldsymbol{X}(i,j)$ denotes the $(i,j)$-th entry.

Below, we describe in detail a single forward pass of a Transformer:

- **Embedding:** A Transformer first maps each token ID to a token embedding vector of dimension $d$. This is done by an embedding layer that applies a pointwise lookup-table function $g : [|\Sigma|] \to \mathbb{R}^d$ to each token ID. In some architectures, positional information is incorporated at the embedding stage by adding a positional encoding (PE) to the embedding vectors, such as the sinusoidal PE in the vanilla Transformer (Vaswani et al., 2017) or the learned PE in BERT (Devlin et al., 2019), to make the model aware of token positions. In other cases, positional information is introduced within self-attention (described later), for example via RoPE (Su et al., 2024), which is used by many LLMs. In summary, the embedding layer can be formulated as a mapping $\mathrm{emb} : [|\Sigma|]^N \times \mathbb{N}^N \to \mathbb{R}^{N \times d}$, which maps at most $N$ token IDs (where $N$ is the context window length, i.e., the maximum input length accepted by the Transformer) together with their position indices to an embedding matrix.

- **Multi-head masked self-attention decoding:** The self-attention mechanism is one of the most fundamental components of Transformers. First, in the $\ell$-th self-attention decoding layer, we apply linear maps to obtain the query, key, and value matrices $\boldsymbol{Q}^{(\ell)}, \boldsymbol{K}^{(\ell)}, \boldsymbol{V}^{(\ell)} \in \mathbb{R}^{N \times d}$, namely $\boldsymbol{Q}^{(\ell)} = \boldsymbol{X}^{(\ell)} \boldsymbol{W}_Q^{(\ell)}$, $\boldsymbol{K}^{(\ell)} = \boldsymbol{X}^{(\ell)} \boldsymbol{W}_K^{(\ell)}$, and $\boldsymbol{V}^{(\ell)} = \boldsymbol{X}^{(\ell)} \boldsymbol{W}_V^{(\ell)}$, where $\boldsymbol{X}^{(\ell)} \in \mathbb{R}^{N \times d}$ is the input embedding matrix and $\boldsymbol{W}_Q^{(\ell)}, \boldsymbol{W}_K^{(\ell)}, \boldsymbol{W}_V^{(\ell)} \in \mathbb{R}^{d \times d}$ are weight matrices. For multi-head masked self-attention, we split each matrix along the embedding dimension into $H$ submatrices $\left\{ \boldsymbol{Q}_h^{(\ell)}, \boldsymbol{K}_h^{(\ell)}, \boldsymbol{V}_h^{(\ell)} \in \mathbb{R}^{N \times d_h} : h \in [H] \right\}$ such that $\sum_{h \in [H]} d_h = d$, and assign them to $H$ attention heads. On each head, the masked self-attention output is computed as

$$\tilde{\boldsymbol{X}}_h^{(\ell)} = \mathrm{softmax}\left( \frac{\boldsymbol{Q}_h^{(\ell)} (\boldsymbol{K}_h^{(\ell)})^\top}{\sqrt{d_h}} + \boldsymbol{M} \right) \boldsymbol{V}_h^{(\ell)}, \tag{1}$$

where $\boldsymbol{M} \in \{0, -\infty\}^{N \times N}$ is the causal mask, defined by $\boldsymbol{M}(i,j) = -\infty$ if and only if $i < j$. We then concatenate $\left\{ \tilde{\boldsymbol{X}}_h^{(\ell)} : h \in [H] \right\}$ to recover the original shape, add the result to the input $\boldsymbol{X}^{(\ell)}$ via a residual connection, and feed it into an MLP (also with a residual connection) to obtain the layer output $\boldsymbol{X}^{(\ell+1)}$.

- **Classification head:** After the final decoding layer, we map the output matrix $\boldsymbol{X}^{(L)} \in \mathbb{R}^{N \times d}$ through a linear projection $\boldsymbol{W} \in \mathbb{R}^{d \times |\Sigma|}$ to obtain the token logits at each position, i.e., $\boldsymbol{Y} = \boldsymbol{X}^{(L)} \boldsymbol{W} \in \mathbb{R}^{N \times |\Sigma|}$.

## B. Proof of Lemma 5.3

**Lemma B.1.** *Let $\Sigma$ be a finite alphabet. For any function $f : \Sigma^2 \to \Sigma$, there exists a decoder-only Transformer $T$ such that for every length-2 context $(a, b) \in \Sigma^2$, greedy decoding outputs $f(a, b)$.*

*Proof.* Without loss of generality, assume $\Sigma = \{1, 2, \cdots, K\}$. Let $d_{\mathrm{model}} = K + 2K + K^2$, and partition each token embedding (row) vector $\boldsymbol{x} \in \mathbb{R}^{1 \times d_{\mathrm{model}}}$[5] into three blocks, $\boldsymbol{x} := \left( \boldsymbol{x}^{\mathrm{tok}}; \boldsymbol{x}^{\mathrm{feat}}; \boldsymbol{x}^{\mathrm{pair}} \right)$, with dimensions $K$, $2K$, and $K^2$, respectively.

We construct a one-layer decoder-only Transformer that consists of (i) multi-head self-attention with two heads (denoted $\mathrm{prev}$ and $\mathrm{self}$), each with $d_k = K$, followed by a residual connection, and (ii) a position-wise feed-forward network (FFN),

---

[5]All arithmetic operations in this proof can be carried out to constant precision (with constants depending only on $|\Sigma|$); we use $\mathbb{R}$ only for notational convenience.

followed by a residual connection. We do not use positional encodings, LayerNorm, or dropout. The construction is summarized as follows:

- **Embedding.** For a token $a$, set its embedding to $(\boldsymbol{e}_a; \boldsymbol{0}; \boldsymbol{0})$, where $\boldsymbol{e}_a \in \mathbb{R}^{1 \times K}$ is the one-hot vector corresponding to $a$.

- **Self-attention.** After applying self-attention to the two input tokens, the representation at position 2 has the form $(*; (\boldsymbol{o}_2^{(\text{prev})}, \boldsymbol{o}_2^{(\text{self})}); \boldsymbol{0})$.

- **FFN.** Using the affine map $\boldsymbol{X}\boldsymbol{W}_1 + \boldsymbol{1}\boldsymbol{b}_1^\top$, we first recover $(*; (\boldsymbol{e}_a, \boldsymbol{e}_b); \boldsymbol{0})$ from $(*; (\boldsymbol{o}^{(\text{prev})}, \boldsymbol{o}^{(\text{self})}); \boldsymbol{0})$, and then apply a second affine map to produce $(*; *; \boldsymbol{e}_{(a-1)K+b})$.

- **Output head.** Finally, we map $\boldsymbol{e}_{(a-1)K+b}$ in the pair block to logits using the classification head.

The details are as follows:

**Step 1: Embedding.** Given input tokens $(a, b) \in \Sigma^2$, define the two token representations

$$\boldsymbol{X} := \begin{bmatrix} \boldsymbol{x}_1 \\ \boldsymbol{x}_2 \end{bmatrix} = \begin{bmatrix} \boldsymbol{e}_a & \boldsymbol{0} & \boldsymbol{0} \\ \boldsymbol{e}_b & \boldsymbol{0} & \boldsymbol{0} \end{bmatrix} \in \mathbb{R}^{2 \times d_{\text{model}}},$$

where $\boldsymbol{x}_1 = (\boldsymbol{e}_a; \boldsymbol{0}; \boldsymbol{0})$ and $\boldsymbol{x}_2 = (\boldsymbol{e}_b; \boldsymbol{0}; \boldsymbol{0})$.

**Step 2: Two-Head Self-Attention.** Let the query matrices be

$$\boldsymbol{W}_Q^{(\text{prev})} := \begin{bmatrix} \sqrt{K}(\boldsymbol{J} - \boldsymbol{I}_K) \\ \boldsymbol{0}_{2K \times K} \\ \boldsymbol{0}_{K^2 \times K} \end{bmatrix}, \qquad \boldsymbol{W}_Q^{(\text{self})} := \begin{bmatrix} \sqrt{K}\boldsymbol{I}_K \\ \boldsymbol{0}_{2K \times K} \\ \boldsymbol{0}_{K^2 \times K} \end{bmatrix},$$

and for each head $h \in \{\text{prev}, \text{self}\}$ define

$$\boldsymbol{W}_K^{(h)} = \boldsymbol{W}_V^{(h)} := \begin{bmatrix} \boldsymbol{I}_K \\ \boldsymbol{0}_{2K \times K} \\ \boldsymbol{0}_{K^2 \times K} \end{bmatrix} \in \mathbb{R}^{d_{\text{model}} \times K}.$$

For head $h$, the query at position 2 is

$$\boldsymbol{q}_2^{(h)} := \boldsymbol{x}_2 \boldsymbol{W}_Q^{(h)} = \begin{cases} \sqrt{K}(\boldsymbol{1} - \boldsymbol{e}_b), & h = \text{prev}, \\ \sqrt{K}\boldsymbol{e}_b, & h = \text{self}, \end{cases}$$

and for $t \in \{1, 2\}$ the keys and values satisfy

$$\boldsymbol{k}_t^{(h)} := \boldsymbol{x}_t \boldsymbol{W}_K^{(h)} = \boldsymbol{v}_t^{(h)} := \boldsymbol{x}_t \boldsymbol{W}_V^{(h)} = \begin{cases} \boldsymbol{e}_a, & t = 1, \\ \boldsymbol{e}_b, & t = 2. \end{cases}$$

Define the attention weights

$$\alpha_{2,t}^{(h)} := \frac{\exp\left(\langle \boldsymbol{q}_2^{(h)}, \boldsymbol{k}_t^{(h)} \rangle / \sqrt{K}\right)}{\sum_{u \in \{1,2\}} \exp\left(\langle \boldsymbol{q}_2^{(h)}, \boldsymbol{k}_u^{(h)} \rangle / \sqrt{K}\right)} = \frac{\exp\left(\langle \boldsymbol{q}_2^{(h)}, \boldsymbol{k}_t^{(h)} \rangle / \sqrt{K}\right)}{\exp\left(\langle \boldsymbol{q}_2^{(h)}, \boldsymbol{e}_a \rangle / \sqrt{K}\right) + \exp\left(\langle \boldsymbol{q}_2^{(h)}, \boldsymbol{e}_b \rangle / \sqrt{K}\right)}$$

$$= \begin{cases} \dfrac{\exp\left(\langle \boldsymbol{1} - \boldsymbol{e}_b, \boldsymbol{k}_t^{(h)} \rangle\right)}{\exp\left(\mathbb{I}[a \neq b]\right) + 1}, & h = \text{prev}, \\[3mm] \dfrac{\exp\left(\langle \boldsymbol{e}_b, \boldsymbol{k}_t^{(h)} \rangle\right)}{\exp\left(\mathbb{I}[a = b]\right) + e}, & h = \text{self}. \end{cases}$$

We next compute the head outputs at position 2, denoted $o_2^{(h)} \in \mathbb{R}^{1 \times K}$. Since

$$o_2^{(h)} := \sum_{t \in \{1,2\}} \alpha_{2,t}^{(h)} v_t^{(h)} = \alpha_{2,1}^{(h)} e_a + \alpha_{2,2}^{(h)} e_b,$$

we obtain

$$o_2^{(\text{prev})} = \begin{cases} \frac{e}{e+1} e_a + \frac{1}{e+1} e_b, & a \neq b, \\ \frac{1}{2} e_a + \frac{1}{2} e_b, & a = b, \end{cases} \qquad o_2^{(\text{self})} = \begin{cases} \frac{1}{e+1} e_a + \frac{e}{e+1} e_b, & a \neq b, \\ \frac{1}{2} e_a + \frac{1}{2} e_b, & a = b. \end{cases}$$

Let the concatenated head outputs be

$$\text{Attn}(X) := \begin{bmatrix} o_1^{(\text{prev})} & o_1^{(\text{self})} \\ o_2^{(\text{prev})} & o_2^{(\text{self})} \end{bmatrix} \in \mathbb{R}^{2 \times 2K}.$$

Choose the attention output projection $W_O := \begin{bmatrix} 0_{2K \times K} & I_{2K} & 0_{2K \times K^2} \end{bmatrix} \in \mathbb{R}^{2K \times d_{\text{model}}}$ so that $\text{Attn}(X) W_O$ writes the two $K$-dimensional head outputs into the middle feature block (and zeros elsewhere). Let $Z := X + \text{Attn}(X) W_O$ be the post-attention representation matrix. Then

$$Z(2,:) = \left( e_b; \left( o_2^{(\text{prev})}; o_2^{(\text{self})} \right); 0 \right).$$

**Step 3: The FFN Builds a Pair Indicator.**    A key observation is that $e_a = \frac{e}{e-1} o_2^{(\text{prev})} - \frac{1}{e-1} o_2^{(\text{self})}$ and $e_b = \frac{e}{e-1} o_2^{(\text{self})} - \frac{1}{e-1} o_2^{(\text{prev})}$. These identities follow by inverting the corresponding linear system when $a \neq b$, and one can verify that they also hold for $a = b$.

Let

$$W_A := \begin{bmatrix} 0_{K \times K} & 0_{K \times K} & 0_{K \times K} & 0_{K \times K^2} \\ 0_{K \times K} & \frac{e}{e-1} I_{K \times K} & -\frac{1}{e-1} I_{K \times K} & 0_{K \times K^2} \\ 0_{K \times K} & -\frac{1}{e-1} I_{K \times K} & \frac{e}{e-1} I_{K \times K} & 0_{K \times K^2} \\ 0_{K^2 \times K} & 0_{K^2 \times K} & 0_{K^2 \times K} & 0_{K^2 \times K^2} \end{bmatrix} \in \mathbb{R}^{d_{\text{model}} \times d_{\text{model}}},$$

so that $(Z W_A)(2,:) = (0; (e_a; e_b); 0)$.

Next, let

$$W_B := \begin{bmatrix} 0 & 0 & \cdots & 0 & 0 & \cdots & 0 & \cdots & 0 \\ e_1^\top & e_1^\top & \cdots & e_1^\top & e_2^\top & \cdots & e_K^\top & \cdots & e_K^\top \\ e_1^\top & e_2^\top & \cdots & e_K^\top & e_1^\top & \cdots & e_1^\top & \cdots & e_K^\top \\ 0 & 0 & \cdots & 0 & 0 & \cdots & 0 & \cdots & 0 \end{bmatrix} \in \mathbb{R}^{d_{\text{model}} \times K^2}.$$

Equivalently, the $((i-1)K + j)$-th column of $W_B$ is $(0; e_i; e_j; 0)^\top \in \mathbb{R}^{d_{\text{model}}}$, and hence for any row vector $x \in \mathbb{R}^{1 \times d_{\text{model}}}$, $(x W_B)((i-1)K + j) = x(K+i) + x(2K+j)$.

Let $W_1 := W_A W_B$ and $b_1 := -1$. Then $(\text{ReLU}(Z W_1 + 1 b_1^\top))(2,:) = e_{(a-1)K+b} \in \mathbb{R}^{1 \times K^2}$.

Finally, choose the second-layer parameters as $W_2 := \begin{bmatrix} 0_{K^2 \times K} & 0_{K^2 \times 2K} & I_{K^2} \end{bmatrix} \in \mathbb{R}^{K^2 \times d_{\text{model}}}$ and $b_2 := 0$. Let

$$X^{(1)} := Z + \left( \text{ReLU}(Z W_1 + 1 b_1^\top) W_2 + 1 b_2^\top \right)$$

denote the FFN output with the residual connection. Then

$$X^{(1)}(2,:) = \left( e_b; \left( o_2^{(\text{prev})}; o_2^{(\text{self})} \right); e_{(a-1)K+b} \right).$$

**Step 4: Output.**    Define $W \in \mathbb{R}^{d_{\text{model}} \times K}$ such that $W(i,j) = 0$ for $i \in \{1, 2, \cdots, 3K\}$ and $j \in \{1, 2, \cdots, K\}$, and $W(3K + (i-1)K + j, k) = \mathbb{I}[f(i,j) = k]$ for $i, j, k \in \{1, 2, \cdots, K\}$. Let $Y := X^{(1)} W$. Then for any $k \in \{1, 2, \cdots, K\}$,

$$Y(2, k) = \sum_{\substack{(i,j) \in \{1, \cdots, K\}^2 : \\ f(i,j) = k}} X^{(1)}(2, 3K + (i-1)K + j) = e_{f(a,b)}(k).$$

Therefore, the greedy decoding output $\arg \max Y(2,:)$ is unique and equals $f(a,b)$.    $\square$

## C. Discussion on Modifying the Context Manager to Simulate Next-Two-Tokens Decoding

In this appendix, we show how to simulate a $(2, 2)$-restricted system while preserving the interface in which each Transformer call decodes a single token. Let $\Sigma$ be a finite alphabet, and let `<1>`, `<2>` $\notin \Sigma$ be two control tokens. Define the extended alphabet $\bar{\Sigma} := \Sigma \cup \{$`<1>`, `<2>`$\}$. Assume that for every function $f : \Sigma^2 \to \Sigma^2$, writing $f(a, b) = (f_1(a, b), f_2(a, b))$, there exists a context-window-3 Transformer over $\bar{\Sigma}$ such that for all $(a, b) \in \Sigma^2$, greedy decoding on input $($`<1>`$, a, b)$ outputs $f_1(a, b)$, and greedy decoding on input $($`<2>`$, a, b)$ outputs $f_2(a, b)$. Under this assumption, we can simulate any $(2, 2)$-restricted system.

Concretely, consider a $(2, 2)$-restricted system specified by a function $M : \Sigma^2 \to \Sigma^2$.[6] Applying the assumption with $f \equiv M$, we obtain a Transformer $T$ that produces the two components of $M(a, b)$ via the control tokens `<1>` and `<2>`. Now consider step $t$ of the simulated $(2, 2)$-restricted system, where the current window content is $(a, b)$. The modified context manager first feeds $($`<1>`$, a, b)$ to $T$ and appends the decoded token to the end of the current sequence. It then feeds $($`<2>`$, a, b)$ to $T$ and appends the decoded token as well. Finally, it deletes the first token of the current sequence, thereby shifting the window forward by one position. This procedure exactly reproduces one step of the $(2, 2)$-restricted system using two single-token decoding calls. Therefore, it suffices to establish that the assumption above holds. We note that this follows by a direct extension of Lemma 5.3 to the context-window-3 setting, and we omit the details.

---

[6]Schuurmans et al. (Schuurmans et al., 2024) require the output length of $M$ to be at most 2. If $M$ outputs the empty string, we treat it as $(\epsilon, \epsilon)$. If $M$ outputs a single token $a$, we treat it as $(a, \epsilon)$, where $\epsilon$ is a special empty symbol that is ignored when appending.

