# OpenReview forum: "Position: The Turing-Completeness of Autoregressive Transformers Relies Heavily on Context Management"
_ICML.cc/2026/Position_Paper_Track — ICML 2026 Position Paper Track regular_

### Official Review · Reviewer_joEN · 2026-02-24

**Significance:** 3
**Argument Clarity:** 3
**Rating:** 5
**Confidence:** 2

**Questions:**

See opportunities for improvement. My questions may stray from the core of the paper, the authors do not need to answer all of them.

**Alternative Views Section:**

Yes

**Compliance With Llm Reviewing Policy A Conservative:**

Affirmed.

**Discussion Potential:**

3

**Paper Summary:**

This paper argues that the Turing-Completeness of real-world autoregressive Transformers relies heavily on context management. It distinguishes two families of Transformers, i.e., fixed-system and scaling family. A clear definition is given to fixed-system Transformers, which match Transformer models used in practice. The Turing-Completeness results for Transformers are reviewed, and the authors show that the results usually assume scaling family Transformers that do not match practical models. The authors also show that the Turing-Completeness of fixed-system Transformers depends heavily on the assumption of their context managers.

**Position:**

Yes

**Position In Title:**

Yes

**Related Work:**

4

**Strengths And Weaknesses:**

I am not familiar with computation theory. I think the strength of the paper include:

S1: A clear definition is given to fixed-system Transformers, and the distinctions between fixed-system Transformers and scaling family Transformers are expressed clearly.

S2: The Turing-Completeness results for Transformers are summarized.

S3: Different context management methods are discussed along with their influences on the Turing-Completeness of fixed-system Transformers.

I think the paper has the following opportunities for improvement.

O1: Turing-Completeness requires to handle unbounded inputs and outputs. In practice, most problems have bounded inputs and outputs. Will this influence the results for Turing-Completeness?

O2: RAG and external memory are important to LLMs in practice. What are the minimum capacities of a RAG or memory system to make fixed-system Transformers Turing-Complete?

O3: I understand that theory problems are important in themselves. Are there examples that Turing-Completeness results guide practical model designs, e.g., changing a model structure to be Turing-Complete and the changed model really has better performance?

**Support:**

4

---

> ### Author Rebuttal · Authors · 2026-03-31
>
> We sincerely thank you for the careful review, thoughtful comments, and valuable suggestions. We address each point below.
>
> ### **Response to O1.**
>
> Turing-completeness is a **purely theoretical** notion. Practical limitations do not affect the validity of the theoretical result itself. Besides, by "unbounded" we mean arbitrarily long but finite inputs, rather than an infinite context window within a single forward pass. An unbounded input length still allows each individual input instance to be finite, although arbitrarily long. In fact, if an LLM is allowed to manage context using the context-management mechanism introduced in our paper, then the overall system can indeed process inputs of arbitrary length. Of course, this does not mean that the context window of a single forward pass is infinite. Nor do we consider possible performance degradation on long inputs, since that belongs to the study of length generalization rather than the focus of this paper. Finally, If the discussion is instead restricted to bounded input and output lengths, then the sets of possible inputs and outputs are finite. Any mapping from one finite set to another finite set, even if the sets are extremely large, is computable.
>
> ### **Response to O2.**
>
> That is a highly practical question. Our preliminary conjecture is that if memory is treated as a tape with a pointer (independent of the context), and if the Transformer is allowed to generate the operations *\<L\>*, *\<R\>*, *\<Read\>*, and *\<Write\>* to move the memory pointer left or right, read the token at the current pointer position into the current context window as the next token, and write the last non-padding token in the current context window to the pointer position, then the resulting system is Turing-complete. However, as we noted in the Call to Action, Turing-completeness only means that the system can mechanically execute instructions to solve any computable problem. It does not take into account the semantics of natural language. In particular, it does not imply that the system can extract well-defined instructions from natural language, which is inherently ambiguous. Therefore, the fact that a RAG or memory-augmented system is Turing-complete does not mean that it can perform perfect retrieval and reasoning from natural language instructions, nor does it say much about its practical performance or generalization ability.
>
> ### **Response to O3.**
>
> In our understanding, current research on the expressive power of Transformers, where Turing-completeness is one example of expressive power, is mostly constructive. That is, such work typically shows that there exists some choice of parameters, architecture, or other mechanism that can solve a certain class of problems. These results characterize the boundary of what the model can represent. Problems within that boundary may be solvable by the model, although whether training can actually find the required parameters requires more refined tools, such as learnability. By contrast, problems outside that boundary are impossible for the model to solve. To use our paper as an example, we prove that summarization-based context management can compute at most constant-space computable problems. Non-regular context-free languages such as $0^n1^n$, however, are not constant-space computable. Therefore, for any fixed context window length $N$, there must exist some $n$ such that a Transformer with summarization-based context management cannot recognize $0^n1^n$. In other words, the accuracy must eventually drop. Once context management is extended to a stronger form, all we can currently conclude is that the resulting system is capable of computing the target function in principle. Whether such behavior can actually be learned through training, and what performance it would achieve in practice, requires finer theoretical tools that go beyond the scope of our work.
>
> Overall, we are very grateful for these thought-provoking questions and for the reviewer’s comments on our paper. We hope that our responses address the concerns and questions.

---

> > ### Author Rebuttal · Reviewer_joEN · 2026-04-06
> >
> > The author response resolves my concerns point-by-point.

---

### Official Review · Reviewer_aJ7U · 2026-03-12

**Significance:** 3
**Argument Clarity:** 3
**Rating:** 4
**Confidence:** 4

**Questions:**

- Can the authors replace the informal “simple enough context manager” notion with a more standard computational restriction, such as finite-state, log-space, or streaming transducers?

- How robust is Proposition 5.1 to more realistic summarization variants, such as multi-token summaries, recursive summaries, or structured summary buffers?

**Alternative Views Section:**

Yes

**Compliance With Llm Reviewing Policy A Conservative:**

Affirmed.

**Discussion Potential:**

3

**Final Justification:**

The author acknowledges that the elements constituting a “simple enough” context manager may be inherently difficult to fully formalize. I agree with this view. As a position paper, I am inclined to accept it.

**Paper Summary:**

The submission examines the concept of Turing-completeness for autoregressive Transformers under realistic deployment assumptions. This paper's general topic consists of distinguishing a fixed-system regime from a scaling-family regime, and arguing that many prior “Transformers are Turing-complete” results actually belong to the latter rather than the former. The paper formalizes a fixed Transformer system as a triple (T,D,C), where T is a fixed pretrained Transformer with constant context length and finite precision, D is a fixed decoding rule, and C is a context manager that determines which tokens are presented to the model at each autoregressive step. Under this framework, the paper proves that summarization-style context management yields only constant-space computation, whereas appending-style context management is equivalent in power to linear-space Turing machines; richer memory interfaces may recover full Turing-completeness. The central claim is that, for real-world autoregressive Transformer systems, context management is not a peripheral implementation detail but a decisive part of the induced computational model.

**Position:**

Yes

**Position In Title:**

Yes

**Related Work:**

3

**Strengths And Weaknesses:**

**Strengths**
- This paper makes an important conceptual clarification. The distinction between fixed-system and scaling-family settings is crisp, meaningful, and highly relevant for interpreting prior universality claims in the Transformer literature.
- The (T,D,C) formalization is simple but effective. It cleanly separates the pretrained model from the decoding rule and, crucially, from the context-management mechanism, which helps make the discussion much more precise.
- The theoretical separation results are also insightful. In particular, showing that summarization-style management is upper bounded by constant-space computation while appending-style management matches linear-space Turing machines strongly supports the paper’s central thesis that context management substantially changes computational power.

**Weaknesses**
- The restriction placed on the context manager C is intuitive but not yet fully canonical. The paper excludes overly powerful managers to avoid trivial Turing-completeness, but the boundary of what counts as a “simple enough” context manager is still somewhat hand-crafted and may deserve a more principled complexity-theoretic treatment.
- The real-world relevance is argued mostly at a conceptual level. Although the paper emphasizes that deployed LLMs rely on context management, it does not yet provide a fine-grained theory for realistic mechanisms such as hierarchical summarization, retrieval pipelines, or tool-use systems.

**Support:**

3

---

> ### Author Rebuttal · Authors · 2026-03-31
>
> We would like to thank you for the time and effort devoted to reviewing our paper and for the thoughtful comments. We respond below to each of your concerns and questions.
>
> ### **On Weaknesses:**
>
> We acknowledge that we do not currently provide a sufficiently clear definition of what constitutes a "simple enough" context manager. To clarify, our framework allows any system $(T, D, C)$ as a valid object of discussion, and we are **not excluding any context manager**. What we mean is that some context managers $C$ are trivially Turing-complete, in which case $(T, D, C)$ becomes trivially Turing-complete as a whole. These are the cases we do not find interesting. In other words, we regard such context managers as trivial and therefore outside the scope of our discussion, **not as invalid or illegitimate**. More broadly, regarding the notion of an intuitive definition, even for classical models of computation such as Turing machines, the claim that they capture real-world computation, namely the Church-Turing thesis, is not itself fully formalized. Therefore, some concepts may inherently resist complete formalization. That said, we appreciate your comment and agree that greater formal clarity would strengthen the paper.
>
> ### **On the Questions:**
>
> - (i) Yes, we agree and will revise the phrasing. As clarified above, we are not claiming that certain context managers are invalid; rather, some of them are trivially Turing-complete, which in turn makes the entire system trivially Turing-complete and therefore less interesting to study. Accordingly, we would like to replace "simple enough context manager" with "a context manager that is not Turing-complete". As for finite-state, log-space, or streaming transducers, we believe all of these are meaningful alternatives worth investigating. In this paper, summarization-based and appending-based context management are intended only as two representative examples. In particular, summarization-based context management is already widely used in real LLM systems.
>
> - (ii) This is a very insightful observation, and an excellent question. In fact, Proposition 5.1 **can be directly extended to multi-token summaries**, including recursive summaries and structured summaries, as long as all summary tokens produced during the summarization process fit within the context window. This assumption is consistent with practical implementations in systems such as Claude and Cursor, and is therefore reasonable. Under this setting, one only needs to replace "generate one summary token" in the definition of summarization-based context management with "generate summary tokens until either an *\<EOS\>* token is produced or the context-length limit is reached". Since the generated summary remains confined to a fixed-length context window, Proposition 5.1 naturally extends to this setting. Thus, as we stated in Lines 209-210, the assumption of generating a single summary token is only a simplification of the definition. Otherwise, we would need to introduce an additional nested summary-generation phase, which would make the definition less clear.
>
> Overall, we are very grateful for your comments on our paper, and we will revise the wording to make the presentation more rigorous. We hope that our response addresses your concerns and questions.

---

> > ### Author Rebuttal · Reviewer_aJ7U · 2026-04-01
> >
> > I thank the author for their response. I decide to maintain my score.

---

### Official Review · Reviewer_q27X · 2026-03-12

**Significance:** 3
**Argument Clarity:** 3
**Rating:** 5
**Confidence:** 3

**Questions:**

Any thoughts on alternative architectures such as SSMs? I imagine the state size of the SSM would play a similar role to the context length in transfomers, and the "scaling family" idea would also apply?

**Alternative Views Section:**

Yes

**Compliance With Llm Reviewing Policy A Conservative:**

Affirmed.

**Discussion Potential:**

3

**Paper Summary:**

This position paper argues that many results that claim transformers are Turing-Complete rely on strong assumptions, such as: (1) For larger context lengths, can use a larger transformer (the "scaling family" setting), or (2) can use infinite/increasing precision for token embeddings as context length increases. This makes the Turing-Completeness claims misleading.

It argues that the above assumptions don't hold in practice: Transformers have fixed size, precision, and context length, and thus going beyond that context length requires a "context manager" that somehow compresses the context to make room for the remaining tokens. Therefore, studying the computational capabilities of Transformers requires deeply understanding the role of the context manager, which cannot be overlooked.

**Position:**

Yes

**Position In Title:**

Yes

**Related Work:**

3

**Strengths And Weaknesses:**

Strengths
- Understanding the limits of transformers in realistic settings is important, given the widespread use of this technology.
- Context management is an increasingly important area as users have longer and longer interactions with LLMs (e.g., Claude Code). This paper highlights the centrality of this process, which is often overlooked.

Weaknesses:
- The line between position paper and technical paper is a bit blurred, given the existence of real results in the paper.

Note: I am not a practicing theoretician, so am not very familiar with the literature discussed here.

**Support:**

4

---

> ### Author Rebuttal · Authors · 2026-03-31
>
> We would like to thank you for the thoughtful comments and for the recognition of our paper. We would like to address your concerns and questions as follows.
>
> ### **On Weaknesses**
>
> - Boundary Between a Position Paper and a Technical Paper
>
> We appreciate this concern and clarify our intended contribution as follows. The main body of the paper aims to provide an overview of the relevant background and perspectives in the literature. On this basis, we distinguish between claims that are consistent with the definition of Turing completeness and those that are not. Although we do present several derivations and conclusions, their purpose is to illustrate how different context-management methods in a fixed-system setting can lead to different levels of expressive power, thereby supporting our position. We include these derivations to make the paper's position more precise and better grounded.
>
> ### **On Problems**
>
> - About SSMs
>
> Yes, we believe your conjecture is correct. Actually, a systematic treatment of architectures such as SSMs is beyond the scope of the current paper. However, if the assumptions in those settings include assumptions from a scaling-family perspective, such as requiring increasingly high numerical precision as the sequence length grows, then such results still cannot be claimed as proofs of Turing completeness. This is because such assumptions do not align with the definition of Turing completeness, which requires a single fixed model to handle inputs of arbitrary length, with tape symbols drawn from a fixed alphabet that does not expand with input length. Since the focus of this paper is on Transformer-based models, we did not include a discussion of SSMs.
>
> We would like to thank you again for the time and effort you devoted to reviewing our paper. We hope this clarification is helpful.

---

> > ### Author Rebuttal · Reviewer_q27X · 2026-04-05
> >
> > I thank the authors for their response, and will keep my positive score.

---

### Official Review · Reviewer_bo6R · 2026-03-13

**Significance:** 3
**Argument Clarity:** 2
**Rating:** 4
**Confidence:** 2

**Questions:**

1. Is there evidence of people misinterpreting the claim of Turing-completeness of transformers as claimed in the alternative view?
2. Are there alternative, realistic context managers besides the appending and summarization styles that the authors think are particularly interesting for study?

**Alternative Views Section:**

Yes

**Compliance With Llm Reviewing Policy A Conservative:**

Affirmed.

**Discussion Potential:**

3

**Final Justification:**

This paper is well-written, well-evidenced, and interesting for discussion. After the rebuttal, I think the authors' position has been clarified to me, and will hopefully be clarified in a camera-ready revision.

My only reservations in giving a score of 5 are that: this submission still somewhat blurs the line of position paper and research paper, and also that this is not my main area of expertise.

**Paper Summary:**

This paper argues that claims of Turing-completeness of autoregressive transformers are more sensitive than commonly interpreted. This is because such proofs typically rely on idiosyncratic forms of context management (i.e., how tokens are handled after the context window is filled) in a way that is unrealistic to real-world implementation. The paper reviews some basics of computability theory, proposes two more "realistic" forms of context management, and proceeds with a brief literature review which illustrates most existing proofs of transformer Turing-completeness require growing contexts or growing numerical precision. The computational power of more realistic context management is then studied.

**Position:**

Yes

**Position In Title:**

Yes

**Related Work:**

3

**Strengths And Weaknesses:**

### Strengths

- The paper is, overall, clearly written. The exposition is self-contained and approachable, and the arguments intuitive.
- The topic itself is extremely relevant, and clearly argues that what is being studied in theory is not what is being used in practice. In principle, it is thus a topic ripe for discussion.
- The argument that existing proofs of Turing-completeness often rely on a "scaling" assumption is well-evidence by a literature review. The technical contents of Section 5 appear also appear correct, but are outside of my direct expertise.

### Weaknesses

- This submission blurs the lines between a position paper and research paper, in my opinion. The "alternative view" appears early on, and principally identifies a gap in the current literature pertaining to a particular research question (the complexity of fixed transformers with different context management strategies). The corresponding research question is then studied, with several technical results derived.
- I'm not sure the final "call to action" has much to do with the overall position. This call to action states "we should develop more suitable notions for characterizing the intelligence of LLMs and other models"; this is intuitive, and similar notions have been argued for the relevance of, e.g., information theory in "intelligence" [1], but this isn't obviously tied to the argument that a transformer's Turing-completeness relies on its context management.
- It's unclear to me the extent to which researchers believe the alternative claim "autoregressive transformers are Turing-complete," where "autoregressive transformer" is a "fixed transformer" in this paper's lingo. It's clear that some modification is required for Turing-completeness (whether this be in the decoding mechanism, infinite precision, read/write memory, etc.), and that saying "transformers are Turing-complete" is plausibly shorthand for "a modified transformer with the same autoregressive self-attention-like mechanism is Turing-complete."
  - It's also unclear that some references are claiming this. As a stark example, Malach is claiming in [2] that auto-regressive next-token predictors are universal learners; indeed, a large part of its novelty and claims are that their results hold for non-transformer (and even linear) models.


[1] Finzi, M., Qiu, S., Jiang, Y., Izmailov, P., Kolter, J. Z., & Wilson, A. G. (2026). From Entropy to Epiplexity: Rethinking Information for Computationally Bounded Intelligence. arXiv preprint arXiv:2601.03220.
[2] Malach, E. (2024, July). Auto-Regressive Next-Token Predictors are Universal Learners. In International Conference on Machine Learning (pp. 34417-34431). PMLR.

**Support:**

4

---

> ### Author Rebuttal · Authors · 2026-03-31
>
> We thank you for your thoughtful comments and address your concerns below.
>
> ### **For the third concern**
>
> We agree that the community may have in mind the claim that "a modified Transformer (e.g., with a different decoding mechanism or infinite-precision inputs) is Turing complete (TC)". What we've found, however, is that most existing works claiming that "Transformers are TC" **actually prove something different**: only a family of Transformers with context-window lengths 100, 1k, 10k, 100k, and so on can, when infinitely many taken together, achieve Turing completeness. The gap between these two statements is precisely the community's **misunderstanding**, and is exactly the distinction we aim to clarify.
>
> We believe there is broad agreement that, as a machine-learning model, a key feature of the Transformer is its **fixed maximum context-window length**. This is what makes training practical and ensures that a single inference pass finishes in bounded time. In fact, all the examples listed in Section 4.2 establish Turing completeness only by combining infinitely many of Transformers with different context-window lengths. We now list several pieces of evidence below.
>
> - [1] In Eq. 13, it's implicitly assumed that at decoding step $t$, the entire sequence fits into the Transformer's context window. This means that, from a complexity-theoretic perspective, handling sequences at different decoding steps requires a family of Transformers rather than one fixed model. Indeed, at step $t$, the sequence has length $n+t$; since $t$ is variable, the context-window size must vary as well.
>
> - [2] Eq. 1-5 make the same implicit assumption that the entire sequence fits into the context window.
>
> - [3] At the top of page 3, the paper treats a softmax-attention Transformer as a map $T:\Sigma^* \to \Sigma^*$. This again assumes that the entire sequence fita into the context window.
>
> - [4] Table 1 shows that, for an input of length $n$, the paper constructs a Transformer with window length $s(n)\ge n$. Thus, different input lengths $n$ require different Transformers.
>
> As for Malach's paper, although it also considers linear AR next-token predictors, the quantity $\hat{h}(\boldsymbol{x}, \boldsymbol{z}_{<t})$ in Def. 3.2 likewise implicitly assumes that the predictor can access the entire prefix. Although there is an upper bound $T$, it is not constant; it varies. In the relevant sense, this still requires a family of predictors whose context-window lengths are $n+t$. For this reason, we also include this paper in our discussion.
>
> An analogy may further help clarify the point. When we say that an idealized computer is TC, we mean that one machine can process inputs of arbitrary length. We don't mean that one machine handles short inputs, a larger machine handles longer inputs, and so on. Likewise, if we want to say that Transformers are TC, the claim should apply to a single Transformer, not to a collection of models with context windows of length 100, 1k, 10k, and so on. This distinction is the central point of our paper. In short, the papers above prove only that **infinite many Transformers, taken together**, is TC; they **don't prove that a modified Transformer** is TC. These aren't the same claim, and conflating them is exactly the community's misunderstanding we seek to clarify. The statement in your comment reflects precisely this conflation.
>
> ### **For other concerns**
>
> - Regarding your comment that our paper blurs the boundary between a position paper and a technical paper, we view it as fundamentally a position paper supported by technical analysis. The main body reviews the background and the different views in the literature. On this basis, we distinguish between claims that are consistent with the definition of Turing completeness and claims that are not. The derivations are included to show how different context-management methods lead to different levels of power, thereby grounding our position. Without them, our argument would lack sufficient support.
>
> - Regarding your concern about our final call to action, we'd also be happy to narrow that discussion. Doing so would not affect the paper's completeness.
>
> - Regarding other realistic context managers, we believe multi-agent systems can be viewed as a more complex form of context management, in which a context manager extracts part of the available context and feeds it to another Transformer for further processing.
>
> Overall, we'd like to thank you again for your review. We hope this rebuttal addresses your concerns, and we'd be very happy to continue the discussion during the discussion phase.
>
> ### **References:**
>
> [1] Pérez et al. On the Turing Completeness of Modern Neural Network Architectures. ICLR 2019.
>
> [2] Qiu et al. Ask, and It Shall Be Given: On the Turing Completeness of Prompting. ICLR 2025.
>
> [3] Jiang et al. Softmax Transformers Are Turing-Complete. arXiv 2025.
>
> [4] Li and Wang. Constant Bit-Size Transformers Are Turing Complete. NeurIPS 2025.

---

> > ### Author Rebuttal · Reviewer_bo6R · 2026-04-03
> >
> > I thank the authors for their detailed rebuttal. I understand that infinite-length prediction windows don't match the "transformer" architecture we use in practice; I suppose my concern was that researchers may not believe the counter to "TC of real-world transformers depends on context management." I.e., the overloaded statement of "transformers are TC" is imprecise but not misunderstood. The authors' rebuttal seems to clarify a more direct position, that "we should study properties of more realistic transformer architectures." This seems worthwhile, and the authors provide ample evidence that the predominant line of study is counter to this.
> >
> > Would the authors be willing to highlight this as a position? (Perhaps not the "main" position, but as an interpretation of the main position).

---

### Decision · Program_Chairs · 2026-04-30

**Decision:**

Accept (regular)

**Comment:**

It is somewhat unclear how to get a paper like this in, which has an important point to make, in any other track.  Clarifying the actual statements about Turing-completeness of transformers, categorizing the literature, and making clear what assumptions are made and how maybe some of the existing claims are a little strong is a service to the community.  I personally found this the most interesting and scientifically valuable of the large set of position papers I meta-reviewed.